

# Performance of risk prediction models for diabetic foot ulcer: a meta-analysis

Panpan Guo[1], Yujie Tu[2], Ruiyan Liu[1,3], Zihui Gao[3], Mengyu Du[3], Yu Fu[1], Ying Wang[4], Shuxun Yan[1] and Xin Shang[1,3]

[1] Department of Endocrinology, The First Affiliated Hospital of Henan University of Chinese Medicine, Zhengzhou, Henan, China
[2] The 154th Hospital, Xinyang, Henan, China
[3] School of First Clinical, Henan University of Chinese Medicine, Zhengzhou, Henan, China
[4] Department of Geriatrics, First Affiliated Hospital of Zhengzhou University, Zhengzhou, Henan, China

## ABSTRACT

**Background:** The number of prediction models for diabetic foot ulcer (DFU) risk is increasing, but their methodological quality and clinical applicability are uncertain. We conducted a systematic review to assess their performance.

**Methods:** We searched PubMed, Cochrane Library, and Embase databases up to 10 February 2024 and extracted relevant information from selected prediction models. The Prediction model Risk Of Bias ASsessment Tool (PROBAST) checklist was used to assess bias risk and applicability. All statistical analyses were conducted in Stata 14.0.

**Results:** Initially, 13,562 studies were retrieved, leading to the inclusion of five development and five validation models from eight studies. DFU incidence ranged from 6% to 16.8%, with age and hemoglobin A1C (HbA1c) commonly used as predictive factors. All included studies had a high risk of bias, mainly due to disparities in population characteristics and methodology. In the meta-analysis, we observed area under the curve (AUC) values of 0.78 (95% CI [0.69–0.89]) for development models and 0.84 (95% CI [0.79–0.90]) for validation models.

**Conclusion:** DFU risk prediction models show good overall accuracy, but there is a risk of bias. Adherence to the PROBAST checklist is crucial for improving their clinical applicability.

Corresponding authors
Shuxun Yan, ysx982001@163.com
Xin Shang, 690379010@qq.com

## BACKGROUND

Background diabetic foot ulcer (DFU) is a severe complication of diabetes arising from neuropathy (*Armstrong et al., 2023*). Globally, about 18.6 million people suffer from DFU annually (*Zhang et al., 2020*), with alarming 5-year mortality rates of 30% for DFU patients and over 70% for amputees (*Armstrong et al., 2020*). These figures significantly impact both life expectancy and quality of life.

In diabetes progression, fluctuations in blood sugar levels can lead to skin ulcers on the lower limbs, triggering inflammation and infection and worsening DFU severity (*Armstrong, Boulton & Bus, 2017*) Hospitalisation or surgical amputation may be

necessary for management, (*Kerr, Rayman & Jeffcoate, 2014*) with over half of DFUs prone to infection and around 20% of severe cases leading to amputation (*Lipsky et al., 2012*). Additionally, DFU patients face a 2.5 times higher 5-year mortality risk compared to non-DFU patients (*Prompers et al., 2007*).

The primary cause of DFU is blood sugar level fluctuations, compounded by risk factors like diabetic peripheral neuropathy (DPN), previous ulcers, foot deformities, or peripheral vascular disease (*Abbott et al., 2002*). Current management focuses on regulating blood sugar, addressing underlying issues, controlling infections, and resorting to surgery when necessary to minimise amputation risks. Despite successful symptom management, recurrence is common post-resolution. Treating diabetes foot complications carries a substantial financial burden, surpassing that of managing common cancers and straining healthcare systems long-term (*Wang et al., 2023*).

Thus, urgent action is needed to mitigate risk factors and develop strategies for DFU prevention. Research suggests that patients receiving clinical care the year before ulcer development have lower amputation risks (*Hinchliffe et al., 2016*). Predictive models incorporating multiple variables enable precise forecasting, empowering proactive intervention to reduce disability rates and amputation risks.

In recent years, there has been a significant increase in the development of DFU risk prediction models. However, these models' methodological quality and predictive accuracy need further evaluation to enhance their clinical relevance. Therefore, this study aims to conduct a comprehensive screening and systematic review of existing DFU risk prediction models, providing up-to-date evidence to support clinical implementation.

## METHODS

This systematic review adhered to the Preferred Reporting Items for Systematic Reviews and Meta-Analyses guidelines (*Snell et al., 2023*; *Tugwell & Tovey, 2021*). The study protocol was registered on PROSPERO (registration number: CRD42023484409).

### Search strategy

A systematic search was performed across multiple databases, including PubMed, Cochrane Library, and EMBASE, spanning from their inception until 10 February 2024. The search strategy involved combining medical subject headings terms and keywords without limiting the language. The keywords utilised include 'Diabetes Mellitus', 'Diabet', 'Prediction model', 'Prognostic model', and 'risk prediction', along with their respective variations (Appendix 1). Additionally, we manually searched through reference lists and relevant systematic reviews to find any possible studies that could be included in the review.

### Inclusion criteria

- Research focused on developing or validating risk prediction models specifically for DFUs
- Diagnostic models aimed at predicting the occurrence or progression of diabetes foot disease
- Studies where the outcome variable is explicitly defined as diabetes foot disease

## Exclusion criteria

- Studies investigating prognosis or other non-diagnostic models
- Studies incorporating an insufficient number of predictive factors (less than two)
- Publications not available in English
- Research solely centred on genetic or biomarker studies as predictive factors
- Conference abstracts, study protocols, duplicate publications, and studies that did not report the desired outcomes were excluded.

## Study selection and screening

Using NoteExpress software for filtering, the literature screening process was conducted independently by two authors (XS and PPG). Initially, duplicate studies were removed, followed by screening the remaining literature based on their titles and abstracts to identify eligible articles. Subsequently, the full text of the remaining articles was meticulously reviewed to determine final inclusion or exclusion based on the predefined criteria. Additionally, references cited within the included articles were examined to ensure the comprehensive identification of relevant studies. In instances of discordance in research selections, the third reviewer (SXY) engaged in discussions to achieve consensus.

## Data extraction

The data collection process involved two reviewers independently gathering relevant information. Basic information included details such as author, publication year, research design, participants, data source, and sample size. Model information included details such as variable selection method, model development method, model validation type, model performance measures, method for processing continuous variables, final predictors used in the model, and form in which the model was presented. Following data extraction, a third reviewer (SXY) validated the collected information. Any disparities were resolved through discussions among the three researchers to ensure consensus.

## Risk of bias and applicability assessment

The bias risk and applicability of each included study were assessed independently by two authors (PPG and XS) using the Prediction Model Risk of Bias Assessment Tool (PROBAST) (*Moons et al., 2019*). In cases of discrepancies between the two authors' assessments, mutual agreement was sought, and if consensus could not be reached, a third reviewer (SXY) was consulted to make a final decision. This tool evaluates the potential risk of bias and applicability across four domains: research subjects, predictive factors, outcomes, and analysis, utilising 20 signal questions. Each domain is evaluated as high, low, or unclear risk. Additionally, applicability assessment covers three areas: participants, predictors, and outcomes, following similar evaluation rules and procedures as the bias risk assessment (*Tan et al., 2023*; *van Beek et al., 2021*).

## Data synthesis and statistical analysis

In this study, the area under the curve (AUC) was computed as the effect measure for model discrimination. To assess heterogeneity, the 95% prediction interval was calculated. Heterogeneity was further evaluated using the $\chi^2$-test and $I^2$-values (*Higgins & Thompson, 2002*). In line with recommendations for high-quality research (*Damen et al., 2023*; *Fu et al., 2024*), a fixed-effects model was employed when $I^2$ was less than or equal to 50% and the *p*-value was greater than 0.1, indicating low heterogeneity (*Qu et al., 2022*). Conversely, a random-effects model was utilised when $I^2$ exceeded 50%, indicating significant heterogeneity (*Higgins et al., 2003*). Sensitivity analysis was conducted to ensure the robustness of the overall findings. Additionally, publication biases were evaluated using funnel plots and Egger's regression test (*Irwig et al., 1998*). All statistical analyses were performed using Stata 14.0 software.

# RESULTS

## Study selection

Overall, 13,562 records were identified through the initial literature search. Of these, 3,063 duplicates were removed, leaving 10,499 unique records. Subsequently, based on the evaluation of titles and abstracts, 10,455 records were excluded. A total of 44 full-text articles were assessed for eligibility. Among these, 32 studies were excluded, as they focused on prognostic models rather than predictive models. Additionally, five studies were excluded, as they contained fewer than two predictive factors, and four studies did not establish prediction models. Ultimately, eight studies met the inclusion criteria and were included in this study (Fig. 1).

## Study characteristics

The studies included in the review spanned publication years from 2006 to 2024. Among them, six were conducted in China, one in the United States, and one in the United Kingdom. Regarding study design, two were prospective studies, four were retrospective studies (including two multicentre studies), one was a retrospective case-control study, and one was a meta-analysis based on cohort studies. The sample sizes across these studies varied, ranging from 299 to 46,521 individuals (Table 1).

Table 2 provides detailed information regarding the predictive models employed in the included studies. Among these studies, six utilised logistic regression analysis to establish predictive models. Notably, in the study by *Boyko et al. (2006)* modelling methods such as risk scoring systems and Cox proportional risk models were also employed. The most frequently utilised predictive factors across the studies were age and hemoglobin A1C (HbA1c), both appearing in each of the five models. Additionally, smoking and Body Mass Index (BMI) were commonly used in four and three models, respectively. Gender, total cholesterol (TC), low density lipoprotein (LDL), DPN, history of foot ulcers, and absence of monofilament sensing were included in two of the models. Reported AUC or C statistical values ranged from 0.65 to 0.934. Calibration was addressed in seven models, with the Hosmer Lemeshow test being the most frequently utilised method.

 

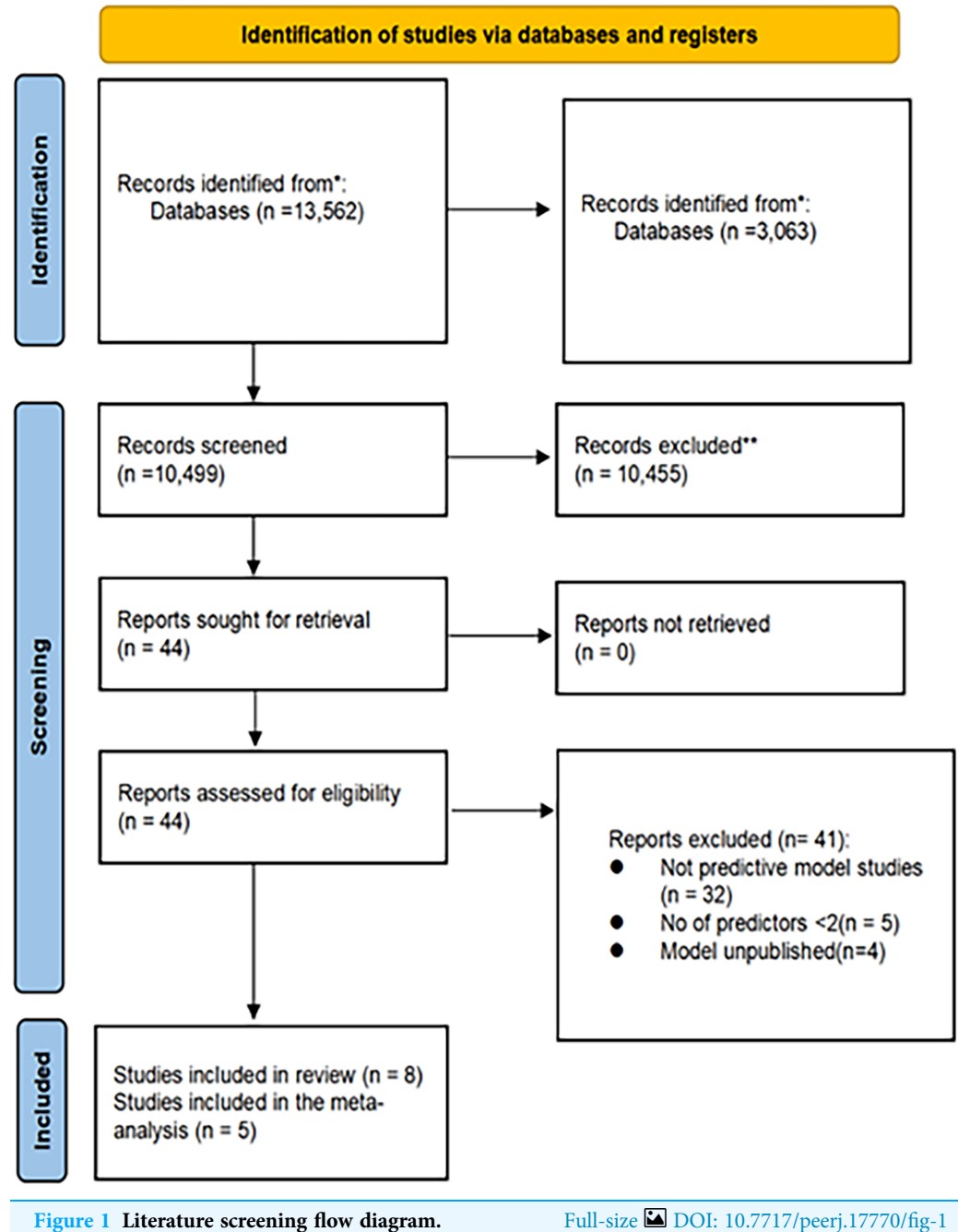

**Figure 1  Literature screening flow diagram.**           

In most of the eight studies, the predictive models underwent either internal or external validation, demonstrating robustness and generalisability. Specifically, three studies underwent external validation, while four studies underwent internal validation. *Peng & Min's (2023)* study stood out, as it underwent both internal and external validation processes. However, two models were not subjected to any validation after their initial development.

**Table 1  Overview of basic data of the included studies.**

| Author (year) | Region | Study design | Participants | Age (SD) (years) | Follow up duration (SD) (year) | Data source | Main outcome | Cases/sample size (%) |
|---|---|---|---|---|---|---|---|---|
| *Heald et al. (2019)* | UK | Retrospective cohort study | Diabetes patients | 16–89 years | 12-year | UK primary care | Foot ulcers occurred | 7% (1,127/17,053) |
| *Lv et al. (2023)* | China | Prospective cohort study | Diabetes patients | 20–80 years 60:3 ± 13:9 years; | 1-year | Department of Endocrinology and Metabolism of a tertiary hospital in Sichuan Province | Foot ulcers | 12.4% (302/2,432) |
| *Chen et al. (2021)* | China | 14 Prospective cohorts and six retrospective cohorts | Patients with type 2 diabetes | 35–80 years | 0.3–19 years | Systematic review and meta-analysis | Diabetic foot ulcer | 6.0% (2,806/46,521) |
| | | Retrospective cohort study | | 56.9 ± 9.8 years | 27 months | Tianjin Medical University Chu Hsien-I Memorial Hospital | Diabetic foot ulcer | 14% (65/465) |
| *Peng & Min (2023)* | China | Retrospective study | T2DM patients | 57.72 ± 12.00 years | — | Wuhan Fourth Hospital and Zhongnan Hospital | Diabetic foot | 17% (84/494) 19.4% (41/211) |
| *Shao et al. (2023)* | China | Retrospective analysis | Diabetic patients | 60 years or older | — | The Department of Orthopedics and Endocrinology, Third Hospital of Shanxi Medical University. | Diabetic foot ulcers | 25.1% (53/211) 31.8% (28/88) |
| *Boyko et al. (2006)* | USA | Prospective data | Diabetic veterans without foot ulcer | 62.4 years | 3.38 years | Veterans Affairs Medical Center | Foot ulcer | 16.8% (216/1,285) |
| *Wang et al. (2022)* | China | Retrospective cohort study | Patients with T2DM | 46.79 ± 2.71 45.12 ± 2.70 | — | The Second Affiliated Hospital of Xi'an Jiaotong University | Diabetic foot | 14.9% (203/1,365) 14.7% (86/585) |
| *Jiang et al. (2022)* | China | Retrospective case-control study | Patients with T2DM | 60.51 (12.7) 63.5 (10.4) | — | Guangxi Medical University First Affiliated Hospital and Wuming Hospital of Guangxi Medical University | Diabetic foot ulcer | 43.3% (369/853) 50% (60/120) |

## Results of quality assessment

We used PROBAST to assess the risk of bias and applicability of all eight included models (Table 3). The assessment of all studies indicating a high risk of bias suggests the presence of methodological issues during either the development or validation phases.

In the participant domain, five studies were identified as having a high risk of bias, primarily attributed to inaccurate data sources (*Boyko et al., 2006*; *Jiang et al., 2022*; *Peng & Min, 2023*; *Shao et al., 2023*; *Wang et al., 2027*). In the predictor domain, one study was deemed to have a high risk of bias due to the inclusion of predictive factors derived from hypotheses (*Heald et al., 2019*). In the outcome domain, four studies were flagged for having a high risk of bias due to the absence of ensuring an appropriate time interval

**Table 2  Overview of the information of the included prediction models.**

| Author (year) | Continuous variable processing method | Variable selection | Model development method | Calibration method | Validation method | Final predictors | Model performance | Model presentation |
|---|---|---|---|---|---|---|---|---|
| Heald et al. (2019) | Categorical variables | — | Single logistic regression model | Hosmer–Lemeshow test. | — | HbA1c, age, absence of monofilament sensation, creatinine level history of stroke | 0.65 (0.62–0.67) | Formula of risk score obtained by regression coefficient of each factor |
| Lv et al. (2023) | Continuous variables | Stepwise regression analysis | Multivariate logistic regression analysis | Brier value | Internal validation | BMI, abnormal foot skin color, foot arterial pulse, callus, history of foot ulcers | Primary cohort: 0.741 (0.7022–0.7799) validation cohort: 0.787 (0.7342–0.8407) | Nomogram and web calculator |
| Chen et al. (2021) | — | — | Scored by its weightings risk scoring system | AUC, calibration plot, Hosmer–Lemeshow test, DCA | Externally validated | Sex, BMI, HbA1c, Smoker, DN, DR, DPN, Intermittent Claudication, Foot care, | Validation cohort: 0.798 (95% CI [0.738–0.858]) | Risk-scoring system based on the systematic review and meta-analysis calculated the score by multiplying the β-coefficient |
| Peng & Min (2023) | — | Forward stepwise regression | Multivariate logistic regression analysis | DCA curve | Internal validation and external validation | Age, smoking history, HbA1C, WBC, LDL-C | Training set 0.827 verification set 0.808 | Nomogram risk prediction model |
| Shao et al. (2023) | — | — | LASSO regression analysis and logistics regression analysis | Calibration diagram | Internal validation | Age, peripheral neuropathy, smoking, high-density cholesterol, lactate dehydrogenase, total serum cholesterol | Training group 0.840 (95% CI [0.779–0.901]) validation group 0.934 (95%CI [0.887–0.981]) | Column line graph prediction models nomogram |
| Boyko et al. (2006) | Continuous variables | Backwards stepwise elimination | Univariate Cox proportional hazards models | — | — | HbA1c, impaired vision, prior foot ulcer, prior amputation, monofilament insensitivity, tinea pedis, onychomycosis | 1 years: 0.81 5 years: 0.76 | Cox proportional hazards modeling |
| Wang et al. (2022) | — | — | Multivariate logistic regression analysis. | Hosmer–Lemeshow test | Internal validation | Age, HbA1c, LDL, TC, smoke, drink | Training cohort: 0.806 (95% CI: [0.775–0.837]) validation cohort 0.857(95% CI: [0.814–0.899]) | Nomogram prediction model |
| Jiang et al. (2022) | Independent variable grouping analysis | Independent variable grouping analysis | Multivariate logistic regression analysis | Consistency index (C index) | External validation | Old age, male gender, BMI, longer duration of diabetes, history of foot disease, cardiac insufficiency, no use of oral hypoglycemic agent (OHA), high white blood cell count, high platelet count, low hemoglobin level, low lymphocyte absolute value | Training cohort 0.89 (0.87–0.91) validation cohort 0.84 (0.77–0.91) | Nomogram |
**Table 3 PROBAST results of the included studies.**

| Author (year) | Study type | ROB | | | | Applicability | | | Overall | |
|---|---|---|---|---|---|---|---|---|---|---|
| | | Participants | Predictors | Outcome | Analysis | Participants | Predictors | Outcome | ROB | Applicability |
| *Heald et al. (2019)* | A | + | — | — | — | + | + | + | — | + |
| *Lv et al. (2023)* | B | + | + | — | — | + | + | + | — | + |
| *Chen et al. (2021)* | B | + | + | + | — | + | + | + | — | + |
| *Peng & Min (2023)* | B | — | + | — | — | + | + | + | — | + |
| *Shao et al. (2023)* | B | — | + | + | — | + | + | + | — | + |
| *Boyko et al. (2006)* | A | — | + | + | — | + | + | + | — | + |
| *Wang et al. (2022)* | B | — | + | + | — | + | + | + | — | + |
| *Jiang et al. (2022)* | B | — | + | — | — | — | + | + | — | — |

between the evaluation of predictive factors and the determination of outcomes (*Heald et al., 2019*; *Jiang et al., 2022*; *Lv et al., 2023*; *Peng & Min, 2023*).

In the analysis domain, all eight studies were assessed to have a high risk of bias. This determination stems from several factors: (1) inadequate sample size that fails to meet established standards; (2) patient follow-up loss exceeding 20%, potentially leading to biased results; (3) inappropriate handling of data complexity, which may compromise the integrity of the analysis; and (4) lack of detailed information regarding participant follow-up, withdrawals, or study terminations, as well as the handling of missing data.

In the assessment of applicability risk, one study was classified as high risk, while the remaining seven studies were deemed low risk. In the participant domain, one study was flagged for high risk, primarily due to a mismatch between the study subjects or environment and the research question (*Jiang et al., 2022*). In both the predictor and outcome domains, all eight studies were classified as low risk. This indicates that the definition of predictive variables and outcome indicators, as well as the timing and system evaluations, were well-aligned with the research objectives, enhancing the applicability and relevance of the predictive models.

## Meta-analysis of development models

Discrepancies exist in the specifics of these models, with incomplete information provided. Only five studies meet the comprehensive criteria. The development model employed a random-effects model to compute the combined AUC, resulting in 0.78 (95% CI [0.69–0.89]) (Fig. 2). Sensitivity analysis of the individual studies revealed no reversal of the pooled-effect size, indicating result robustness (Appendix 2 Figure A). The Egger test yielded a result of 0.364, suggesting no significant evidence of publication bias.

## Meta-analysis of validation models

The validation model utilised a random-effects model to compute the combined AUC, resulting in 0.84 (95% CI [0.79–0.90]) (Fig. 3). The $I^2$ value is 80.7% ($p < 0.001$), indicating notable heterogeneity among the studies. Furthermore, sensitivity analysis confirms result robustness, with no individual studies altering the pooled-effect size (Appendix 2

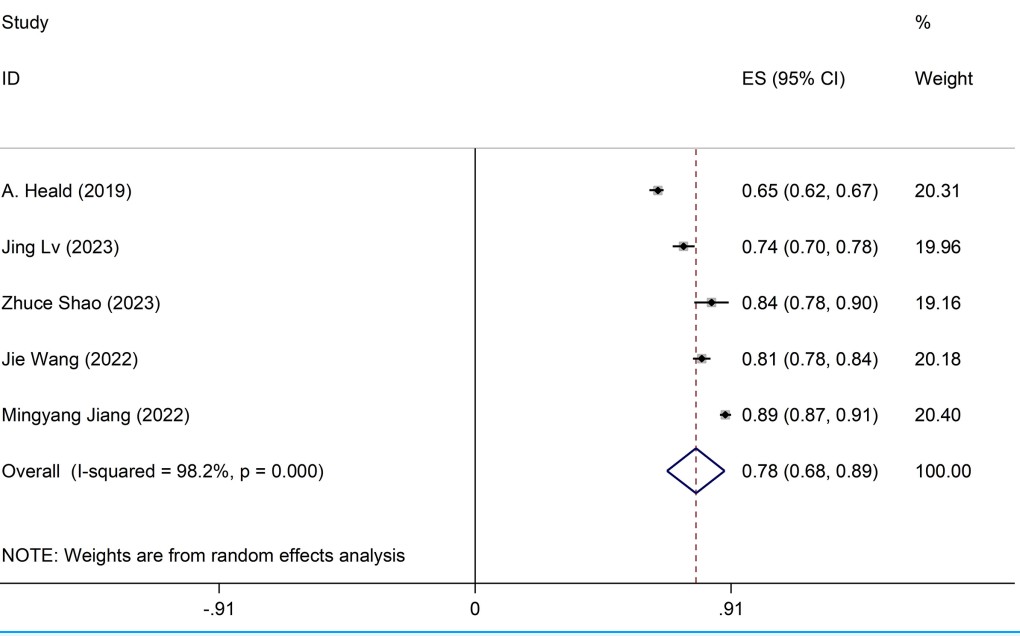

**Figure 2 Forest plot of pooled AUC estimates for development models.**

Study                                                                          %

ID                                                      ES (95% CI)        Weight

Jing Lv (2023)                                          0.79 (0.73, 0.84)  20.21

Dong Chen (2021)                                        0.80 (0.74, 0.86)  19.20

Zhuce Shao (2023)                                       0.93 (0.89, 0.98)  21.13

Jie Wang (2022)                                         0.86 (0.81, 0.90)  21.76

Mingyang Jiang (2022)                                   0.84 (0.77, 0.91)  17.70

Overall  (I-squared = 80.7%, p = 0.000)                 0.84 (0.79, 0.90)  100.00

NOTE: Weights are from random effects analysis

-.981                    0                    .981

**Figure 3 Forest plot of pooled AUC estimates for validation models.**

Figure B). The Egger test yielded a result of 0.21, suggesting no significant evidence of publication bias.

## DISCUSSION

In our meta-analysis evaluating foot ulcer risk prediction models in diabetes patients, we analysed five development and five validation models across eight studies, primarily

involving Chinese patient data, with AUC values ranging from 0.65 to 0.93. Despite their promising predictive capabilities, all studies exhibited a high risk of bias based on the PROBAST checklist, undermining their practical utility. We observed AUC values of 0.78 (95% CI [0.69–0.89]) for development models and 0.84 (95% CI [0.79–0.90]) for validation models, alongside significant heterogeneity likely due to variable population characteristics, predictive factors, and methodologies. To improve the predictive models' utility for assessing foot ulcer risk in diabetes patients, future research should focus on developing new models through larger, rigorously designed studies encompassing multi-centre external validations and enhanced reporting transparency. Such efforts are vital for enabling precise risk assessments and early interventions, ultimately reducing the DFU burden and enhancing patient outcomes.

DFUs often arise due to minor wounds and inflammation resulting from foot care negligence during the course of diabetes. These wounds can lead to foot skin bleeding, persistent non-healing, and, in severe cases, ulceration, inflammation, and infection, causing tissue damage. Around 34% of diabetes patients eventually develop DFUs, with roughly half of these becoming infected, requiring hospitalisation for treatment (*Armstrong, Boulton & Bus, 2017*). Furthermore, 15% to 20% of moderate to severe infections ultimately necessitate lower limb amputation (*Petersen et al., 2022*; *Senneville et al., 2024*). A meta-analysis demonstrated that patients with DFUs have higher all-cause mortality rates compared to those without foot ulcers (*Saluja et al., 2020*). Consequently, the accurate assessment of DFU risk in diabetic patients, along with early detection and intervention, is crucial in reducing the incidence and severity of adverse outcomes.

The frequent occurrence of specific predictive factors in the model holds significant implications for clinical guidance. Age and HbA1c stand out as high-frequency predictors, along with commonly used indicators like smoking and BMI. Age is particularly noteworthy as a risk factor for chronic diabetes complications, especially among the elderly, where the risk significantly increases. This elevated risk in older individuals can be attributed to the progressive nature of diabetes and its associated complications. Studies over 15 years have shown that elderly diabetes patients have a potentially higher incidence rate of DFU, highlighting the importance of age as a key predictive factor in assessing and managing DFU risk (*Tai et al., 2021*).

HbA1c plays a crucial role in assessing the risk of diabetic complications such as foot ulcers. Serving as a marker for long-term glycaemic control, HbA1c reflects average blood glucose levels over 2–3 months (*Dogan et al., 2019*). Many patients struggle to maintain optimal levels, recommended below 6.5% by guidelines (*American Diabetes Association, 2017*). Research such as *Boyko et al.*'s *(2006)* study has shown predictive value of HbA1c in forecasting foot ulcer risks. Higher HbA1c levels correlate with increased complication risks, emphasising the importance of stable control for improving prognosis and preventing adverse outcomes in diabetes patients (*Hasan et al., 2016*).

Obesity and smoking are well-established risk factors for foot ulcers in diabetes (*Tola, Regassa & Ayele, 2021*). Obesity leads to heightened foot pressure in diabetic individuals and is linked to elevated blood lipids, metabolic dysfunction, and inflammation, all contributing to DFU development. Recent research emphasises obesity's impact on DFU

prevalence and incidence. Conversely, regular exercise has been proven beneficial in both preventing and managing DFUs, thereby improving prognosis (*Wang et al., 2022*).

Controlling the smoking risk factor is crucial for enhancing the prevention and treatment of foot ulcers in diabetes patients (*Yang, Rong & Wu, 2022*). Smoking accelerates atherosclerosis, reducing blood circulation and leading to earlier amputations in smokers compared to non-smokers, highlighting the harmful effects on diabetic complications (*Xia et al., 2019*). Research indicates that quitting smoking can enhance amputation-free survival rates in diabetes patients. Furthermore, smoking is associated with an elevated risk of infection with ESKAPE pathogens in DFU (*Li et al., 2022*). Therefore, patients can significantly benefit from preventive measures like smoking cessation, effectively strengthening protection against DFU (*Singh, Armstrong & Lipsky, 2005*).

Certainly, early intervention and stable blood glucose control are crucial in reducing the impact of blood glucose fluctuations on diabetes complications. Lifestyle changes such as regular exercise and quitting smoking are key in delaying diabetic complications and lowering DFU risk. Existing predictive models offer valuable insights for future research, aiding in identifying additional risk factors and developing more comprehensive models. Given the numerous risk factors linked to DFU, early prevention and intervention are vital in mitigating its risks. Taking proactive measures and addressing modifiable risk factors enable healthcare providers to effectively lessen the DFU burden and enhance outcomes for diabetic individuals.

This meta-analysis highlights several important considerations regarding potential limitations. First, the overrepresentation of model studies focused on the Chinese population may introduce regional biases, limiting the generalisability of findings to other geographic areas. To address this, future research should prioritise including more diverse and larger sample sizes, validating across different populations and regions to enhance the robustness and applicability of predictive models. Second, due to data incompleteness and methodological differences, our meta-analysis only included a subset of development and validation models from the identified studies. To mitigate this limitation, future studies should adhere to rigorous methodological standards, follow PROBAST checklist guidelines, and ensure comprehensive reporting for a more accurate synthesis of evidence. Lastly, despite conducting a thorough literature search, there remains a possibility of missing relevant citations, potentially underestimating the total number of developed and validated models. To address this, researchers should continue comprehensive searches across multiple databases and sources, considering systematic review methodologies to minimise the likelihood of overlooking pertinent studies.

## CONCLUSION

DFU risk prediction models generally exhibit good overall predictive accuracy. Nonetheless, there is a notable risk of bias during their development and validation phases. It is vital to improve the calibration performance of existing models, ensuring their suitability for the general population. In future research, priority should be given to assessing model applicability, improve the quality of the model and closely following the PROBAST checklist to enhance clinical relevance and value.

### Funding

This study was supported by the Special Research Project of Traditional Chinese Medicine in Henan Province (2018JDZX018), the Henan Traditional Chinese Medicine Inheritance and Innovation Talent Project (Zhongjing Project) Chinese Medicine Top Talent Project (CZ0237-02), and the Leader of the Young and Middle aged Discipline of Henan Provincial Health Commission. The funders had no role in study design, data collection and analysis, decision to publish, or preparation of the manuscript.

### Grant Disclosures

The following grant information was disclosed by the authors:
Special Research Project of Traditional Chinese Medicine in Henan Province: 2018JDZX018.
Henan Traditional Chinese Medicine Inheritance and Innovation Talent Project (Zhongjing Project) Chinese Medicine Top Talent Project: CZ0237-02.
Leader of the Young and Middle aged Discipline of Henan Provincial Health Commission.

### Competing Interests

The authors declare that they have no competing interests.

### Author Contributions

- Panpan Guo conceived and designed the experiments, performed the experiments, analyzed the data, authored or reviewed drafts of the article, and approved the final draft.
- Yujie Tu performed the experiments, analyzed the data, prepared figures and/or tables, and approved the final draft.
- Ruiyan Liu performed the experiments, analyzed the data, authored or reviewed drafts of the article, and approved the final draft.
- Zihui Gao performed the experiments, prepared figures and/or tables, and approved the final draft.
- Mengyu Du performed the experiments, prepared figures and/or tables, and approved the final draft.
- Yu Fu analyzed the data, authored or reviewed drafts of the article, and approved the final draft.
- Ying Wang analyzed the data, authored or reviewed drafts of the article, and approved the final draft.
- Shuxun Yan analyzed the data, authored or reviewed drafts of the article, and approved the final draft.
- Xin Shang conceived and designed the experiments, performed the experiments, analyzed the data, authored or reviewed drafts of the article, and approved the final draft.

### Data Availability

The raw measurements are available in the Supplemental Files.

## Supplemental Information

Supplemental information for this article can be found online at http://dx.doi.org/10.7717/peerj.17770#supplemental-information.

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
