# Peer review of "Performance of risk prediction models for diabetic foot ulcer: a meta-analysis"

_PeerJ, doi:10.7717/peerj.17770_

## Round 0.1 · original submission · Major Revisions

Dear Authors, As you can see, we had three reviewers with very antagonistic reviews. One of the reviewers suggested the rejection of the manuscript and two other reviewers suggesting the accept of the manuscript with, of course, some minor revisions. Therefore, my decision is MAJOR REVISIONS. Please, try to make all changes posed by reviewers and revise specifically all the points posed by reviewer 1 whose revision, in my view, was exaggerated negative.

Reviewer 1 ·

Basic reporting

English grammar needs improvement

Experimental design

no comment

Validity of the findings

Lack of innovation and novelty

Additional comments

This manuscript summarizes the risk prediction models of diabetic foot ulcer (DFU). But there are many serious issues with the content
1.The manuscript lacks innovation, and many similar articles have been published.(Maldonado-Valer T, etal. Prevalence of diabetic foot. PLoS One),(Guo QJ, etal. [Construction and preliminary validation of a risk prediction models]. Zhonghua Shao Shang Yu Chuang Mian Xiu Fu Za Zhi. ),(Maldonado-Valer T, etal. Prevalence of diabetic foot at risk of ulcer. PLoS One. )
2.There are many English grammar errors in the entire manuscript, which need improvement.
3.The research results are too simplistic, only incorporating AUC. And the conclusion drawn from the DFU risk prediction model shows good overall accuracy, without any value or significance.
4.The study protocol was registered on PROSPERO (registration number: CRD42023484409).This registration number was not found in PROSPERO, is there an error?
5.The inclusion and exclusion criteria were not strictly written according to the PICO principle, and the description was unclear.
6.The heterogeneity analysis I2 of the forest plot results in the entire text is greater than 50%, so the credibility of the results obtained is not sufficient

Reviewer 2 ·

Basic reporting

no comment

Experimental design

no comment

Validity of the findings

no comment

Additional comments

The development and establishment of prediction models are on the rise, yet only a few of these models have undergone validation and seen widespread application. Therefore, assessing the usability of these models and conducting systematic evaluations of predictive models hold great significance for future research endeavors.

This article systematically evaluates predictive models for diabetic foot, encompassing development models and validation models from eight studies. This meta-analysis adheres to established protocols.

Minor problems
Table 2 exhibits some syntax and punctuation issues.
There may be a typographical error in the I² value on line 190.
As mentioned earlier, while this study is valuable, future research efforts should prioritize the development of new models with larger sample sizes, rigorous research designs, multicenter external validation, and heightened reporting transparency. Providing a more detailed elucidation of the significance of this study would further enhance its value.

Reviewer 3 ·

Basic reporting

Compliant with standards

Experimental design

Compliant with standards

Validity of the findings

Compliant with standards

Additional comments

Your paper appears to be well-structured and covers an important topic in diabetes research.But there are still some minor issues
Some sentences are long and could be broken down for better readability. "DFUs often arise due to minor wounds and inflammation resulting from foot care negligence during the course of diabetes" could be split into two sentences for clarity.

Ensure that all technical terms and concepts are accurately represented. For instance, when discussing HbA1c, make sure to provide its full name (hemoglobin A1c) upon the first mention, and explain its significance briefly for readers who may not be familiar with the term.


Check for consistency in terminology and language usage throughout the paper. For example, consider using "foot ulcer" consistently instead of alternating between "foot ulcer" and "DFU" (Diabetic Foot Ulcer) to avoid confusion.


It's good that you address potential limitations of your study, such as regional biases and data incompleteness. Consider expanding on how these limitations might affect the interpretation of your results and what steps could be taken to mitigate them in future research.

Overall, the paper presents valuable insights into the development and validation of foot ulcer risk prediction models in diabetes patients. With some revisions for clarity, accuracy, and completeness, it has the potential to make a significant contribution to the field.

---

## Round 0.2 · accepted · Accept

The authors have addressed all of the reviewers' comments, and all the reviewers are satisfied with the authors' corrections and revisions. Therefore, I am happy to inform you that my editorial decision is to accept the manuscript. Congratulations!

Reviewer 1 ·

Basic reporting

no comment

Experimental design

no comment

Validity of the findings

no comment

Additional comments

I have reviewed the entire article and found that it has been completely revised. I suggest accepting it

Reviewer 2 ·

Basic reporting

no comment

Experimental design

no comment

Validity of the findings

no comment

Additional comments

The innovation, rigor of experimental design, and depth of data analysis of this article have all reached the level of the journal

Reviewer 3 ·

Basic reporting

no comment

Experimental design

no comment

Validity of the findings

no comment

Additional comments

The problem has been solved, and it is recommended to accept it